# Enoxaparin Increases D6 Receptor Expression and Restores Cytoskeleton Organization in Trophoblast Cells from Preeclampsia

**DOI:** 10.3390/cells11132036

**Published:** 2022-06-27

**Authors:** Chiara Tersigni, Giuseppe Maulucci, Roberta Castellani, Giada Bianchetti, Marianna Onori, Rita Franco, Greta Barbaro, Marco De Spirito, Antonio Lanzone, Giovanni Scambia, Nicoletta Di Simone

**Affiliations:** 1U.O.C. di Ostetricia e Patologia Ostetrica, Fondazione Policlinico Universitario A. Gemelli IRCCS, L. go A. Gemelli 8, 00168 Rome, Italy; mariannaonori@gmail.com (M.O.); antonio.lanzone@policlinicogemelli.it (A.L.); 2Dipartimento di Neuroscienze, Sezione di Biofisica, Università Cattolica del Sacro Cuore, L. go Francesco Vito 1, 00168 Rome, Italy; giuseppe.maulucci@unicatt.it (G.M.); giada.bianchetti@unicatt.it (G.B.); marco.despirito@unicatt.it (M.D.S.); 3Fondazione Policlinico Universitario A. Gemelli IRCCS, L. go A. Gemelli 8, 00168 Rome, Italy; 4Istituto di Clinica Ostetrica e Ginecologica, Università Cattolica del Sacro Cuore, L.go Francesco Vito 1, 00168 Rome, Italy; roberta.castellani@unicatt.it (R.C.); rita.f7593@gmail.com (R.F.); greta.barbaro@libero.it (G.B.); giovanni.scambia@policlinicogemelli.it (G.S.); 5U.O.C. di Ginecologia Oncologica, Fondazione Policlinico Universitario A. Gemelli IRCCS, L. go A. Gemelli 8, 00168 Rome, Italy; 6Department of Biomedical Sciences, Humanitas University, Via Rita Levi Montalcini 4, 20072 Pieve Emanuele, Italy; nicoletta.disimone@hunimed.eu; 7IRCCS Humanitas Research Hospital, Via Manzoni 56, 20089 Rozzano, Italy

**Keywords:** heparin, preeclampsia, D6, chemokines, placenta, cytoskeleton

## Abstract

D6 is a scavenger receptor for CC chemokines expressed in the human placenta. It prevents excessive leukocyte tissue infiltration by internalizing chemokines through cytoskeleton-dependent intracellular transport. In preeclampsia (PE), the D6 receptor is overexpressed in trophoblast cells, but functionally impaired, due to cytoskeleton destructuring. Low molecular weight heparin (LMWH) represents a potential treatment for PE based on its anti-thrombotic and anti-inflammatory properties. Here, we investigated the effect of enoxaparin on D6 expression, and cytoskeleton organization primary cytotrophoblast cell cultures were obtained from the placentae of women with PE (*n* = 9) or uncomplicated pregnancy (*n* = 9). We demonstrated that enoxaparin is able to (i) increase D6 expression, and (ii) improve cytoskeletal fiber alignment in trophoblast cells from PE patients.

## 1. Introduction

Preeclampsia (PE) is a pregnancy-specific hypertensive disorder and a major cause of maternal and neonatal morbidity and mortality, complicating 2–5% of all pregnancies worldwide [1,2].

PE is characterized by exaggerated systemic inflammation, involving leukocytes, the clotting and the complement systems, and the endothelium.

Higher circulating levels of pro-inflammatory cytokines and CC chemokines have been shown in PE [3,4,5,6,7], and are responsible, the former, for communication within this inflammatory network, and the latter, for promoting leukocytes recruitment, by activating conventional G protein-coupled receptors [8,9]. Chemokines are also recognized by a set of atypical chemokines receptors (ACRs), such as the D6 decoy receptor, that generate chemokine gradients in tissues and modulate tissue inflammation. D6 binds most inflammatory CC chemokines, and internalizes and targets the ligand for degradation [10,11]. Then, the receptor is free to recycle back to the cell surface with mechanisms that are strictly dependent on cytoskeleton dynamics [12,13,14,15]. Engagement of the D6 receptor by its ligands activates an intracellular signaling pathway, leading to phosphorylation and inactivation of a major actin-depolymerizing factor, the cofilin. This process enables actin network rearrangements that are critically required for the increased abundance of D6 protein on the cell surface, and for its chemokine-scavenging activity [15]. D6 expression has been detected in the human placenta [3,16], particularly in the apical surface of chorionic villous trophoblasts [3]. Its placental expression is functional to the maintenance of a controlled inflammatory placental environment at the maternal–fetal interface [3,17]. Indeed, D6 has been shown in animal models to prevent excessive placenta leukocyte infiltration and inflammation-induced fetal loss [17].

In PE, increased syncytiotrophoblast expression of the D6 decoy receptor has been documented [7]. In contrast, a functional assay of D6 receptor scavenging activity in primary trophoblast cell cultures from PE has shown a significant decrease of exogenous CCL2 scavenging activity [7]. The functional impairment of the D6 receptor was related to a dramatic destructuring of the trophoblast cells’ cytoskeleton, which is crucial for D6 scavenging activity, compatible with cell inflammation and oxidative stress occurring in the syndrome [7].

Low molecular weight heparin (LMWH) is the anticoagulant of choice in pregnancy because it does not cross the placenta and has a favorable maternal safety profile. Because of its ability to prevent placental thrombosis and modulate inflammation, administration in pregnancy of prophylactic doses of LMWH has been proposed to prevent recurrence of PE [18]. Nonetheless, its use with this indication is not currently recommended due to contrasting results coming from clinical studies [19,20,21,22,23].

LMWH has been shown in vitro to enhance B6FS fibrosarcoma cell adhesion and migration through a FAK/actin cytoskeleton axis, particularly by inducing F-actin polymerization and promoting cytoskeleton rearrangement [24].

Based on this evidence, the aim of this pilot study was to investigate the in vitro effect of enoxaparin on D6 decoy receptor function and cytoskeleton organization in trophoblast cells from PE patients.

## 2. Materials and Methods

### 2.1. Patients and Samples

Placentae were collected at the delivery suite of the Gemelli Hospital in Rome, Italy, from 9 women with normal pregnancies undergoing caesarean section for breech presentation, and 9 women with PE, immediately after delivery. PE was defined as new onset of hypertension and substantial proteinuria occurring after the 20th week of gestation [25]. None of the women recruited in this study were in labor at the time of placental sampling. All women who donated their placentae for this study had a singleton pregnancy with no known fetal abnormalities. Women with diabetes, autoimmune diseases, infections, or sepsis were excluded from this study. This study was approved by the Ethics Committee of the Catholic University of Sacred Heart, Rome, Italy, and conducted according to the Declaration of Helsinki. Informed written consent was collected from all participants.

### 2.2. Cell Cultures

Cytotrophoblast cells were isolated from placentae of PE (*n* = 9) and normal pregnancy (*n* = 9), as detailed elsewhere [26]. Cell viability, assessed by trypan blue dye exclusion, was 90%. The purity of the cell preparation was evaluated by immunohistochemical staining for markers of (a) macrophages (3%, determined using a polyclonalanti-a1-chymotrypsin antibody; Dako, SantaBarbara, CA, USA); (b) fibroblasts (2%, determined using a polyclonal anti-vimentin antibody; Lab systems, Helsinki, Finland); and (c) syncytiotrophoblasts (1% determined using a monoclonal antibody against low molecular weight cytokeratins; Lab systems, Chicago, IL, USA). Then, enriched (95%) trophoblast cells were cultured in Dulbecco’s modified Eagle’s medium (DMEM, Sigma-Aldrich, St. Louis, MO, USA) with 10% fetal bovine serum (FBS, Sigma) at 37 °C in 5% CO_2_/95% air for 24 h. Cells were then rinsed twice in PBS to remove detached cells and incubated with fresh medium with or without enoxaparin (1–10 IU/mL, Clexane^®^ Sanofi, Paris, France) for another 24 h.

### 2.3. Western Blotting

Following incubation with enoxaparin, trophoblast cells were detached using trypsin and centrifugation. Cell pellets were resuspended in 2.5 mL of RIPA lysis buffer (30 mM HEPES, pH 7.4, 150 mM NaCl, 1% Nonidet P-40, 0.5% sodium deoxycholate, 0.1% sodium Dodecanese sulfate, 5 mM EDTA; Sigma-Aldrich, St. Louis, MO, USA) containing freshly added protease inhibitors (200 μM pheylmethylsulphonylfluoride and 1 μM leupeptin; Sigma-Aldrich) and homogenized in a 5 mL glass Dounce homogenizer. The homogenate was centrifuged at 700× *g* for 5 min to remove nuclei and unbroken cells, and the resultant supernatant was resuspended in homogenizing buffer and frozen at −80 °C until use. Protein concentration was measured by BCA.

Immunoblotting was performed using standard protocols with trophoblast cell lysates (20 µg) from PE (*n* = 3) and healthy pregnant women (*n* = 3). Briefly, cell lysates were incubated at 4 °C overnight with rat primary anti-human D6 antibody (dilution 1:1000; Santa Cruz Biotechnology, Dallas, TX, USA) and mouse anti-actin antibody (dilution 1:500; Thermo Fisher Scientific, Waltham, MA, USA) as loading control in Tris-buffered saline and 0.05% Tween 20 (TBS-T) containing 1% BSA. After washing, cell lysates were incubated with HRP-conjugated secondary antibody (1:4000; Dako, Glostrup, Denmark) for 1 h at R/T. Immunoblots were treated with an enhanced chemiluminescence detection system (PierceTM, Thermo Fischer Scientific, Waltham, MA, USA) and exposed to Hyperfilm ECL (GE Healthcare Life Sciences, Cleveland, OH, USA).

### 2.4. Confocal Microscopy Analysis of D6 Expression in Trophoblast Cells

Trophoblast cells were seeded on glass bottom dishes (Ibidi, Gräfelfing, Germany, 5 × 10^4^ cell/dish) and treated with medium only or incubated with enoxaparin (10 IU/mL; Clexane^®^ Sanofi, Paris, France) for 24 h at 37 °C. Cells were then rinsed twice in PBS, fixed with 4% PFA for 10 min and incubated for 1 h at R/T with primary anti-D6 antibody (10 μg/mL of rat monoclonal anti-human D6, R&D Systems, Minneapolis, MO, USA) and, then, with secondary goat anti-rat Alexa Fluor 594 conjugated antibody (Life Technologies, Carlsbad, CA, USA) at 1:400 dilutions for 1 h at R/T. All samples were visualized using an inverted confocal microscope (Nikon A1-MP, Tokyo, Japan) equipped with an on-stage incubator (OKOLAB). Images were acquired with a 60× oil-immersion objective (1.4 NA) in 16-bit, with a 1024 × 1024 pixels resolution. Internal photon multiplier gain was kept constant at the same operating voltage during the experiment. The Alexa Fluor 594 conjugated antibody was excited with a single-photon laser (excitation wavelength: 488 nm) and fluorescence was collected in the emission range 500–525 nm.

To evaluate intracellular fluorescence intensity, images were analyzed with the open-source software ImageJ version 1.38 (NIH, Bethesda, MD, USA). After the isolation of cells from the background, through a pixel-classification workflow [27], fluorescence emission intensity was determined within multiple regions of interest (ROI) using the Find Maxima tool, and retrieved values were then normalized with respect to the laser power used for different acquisitions.

### 2.5. F-Actin Immunofluorescent Staining

To investigate cytoskeleton organization, trophoblast cells from PE (*n* = 9) or control placentae (*n* = 9) were seeded on glass bottom dishes (Ibidi, Gräfelfing, Germany) at a concentration of 5 × 10^4^ cell/dish in DMEM with 10% FBS and incubated with medium only or with enoxaparin (10 IU/mL; Clexane^®^ Sanofi, Paris, France) for 24 h at 37 °C. Cells were then rinsed twice in PBS, fixed with 4% PFA for 10 min at R/T, and permeabilized with 0.1% Triton-X for 5 min. Next, cells were incubated with Phalloidin Fluorescein Isothiocyanate labeled according to manufacturer’s instructions for 30 min at R/T (Sigma-Aldrich, St. Louis, MO, USA). F-actin fibers were visualized by inverted confocal microscope as previously described. Phalloidin was excited with a single-photon laser (excitation wavelength: 488 nm) and fluorescence was collected with an emission filter 525/50 nm.

### 2.6. Measure of Cytoskeletal Fiber Alignment

To measure cytoskeletal fiber alignment, we performed the Fast Fourier Transform (FFT) with the open-source software ImageJ (NIH). If the image in the real space contains aligned and organized features, the corresponding 2D FFT shows an elongated (elliptical) shape in the frequency domain. Conversely, an isotropic signal results in a rounder shape [28]. Measuring the eccentricity of the ellipse in the frequency domain can thus provide a good indicator of cytoskeleton organization. To do this, several squared ROI (100 × 100 pixels) were selected in correspondence of the inner part of the cells and the FFT was calculated. A threshold was then applied to the frequency domain image to isolate the information coming from structures of interest in the ROI, and the features of the obtained shape were measured. The eccentricity *e* was calculated according to the following formula:$$e=\frac{c}{a}$$
where $c=\sqrt{a^{2}-b^{2}}$, *a* being the major semi-axis and *b* the minor one. Eccentricity values range from 0 (isotropic signal, round shape) to 1 (organized features, elongated shape).

### 2.7. Statistical Analysis

Means and standard deviation (SD) or medians and minimum–maximum ranges were used to describe quantitative variables, whereas absolute and relative frequencies were employed for categorical ones. Data were analyzed using one-way analysis of variance (ANOVA) followed by a post-hoc test (Bonferroni test). The results are shown as the mean ± standard error (SE) or median with related confidence interval. For all analyses, *p* < 0.05 was considered significant.

## 3. Results

### 3.1. Study Population

The clinical characteristics of PE patients and healthy pregnant women enrolled in this study are reported in Table 1. Pre-eclamptic women showed higher body mass index (BMI) (*p* < 0.001), lower gestational age at delivery (*p* < 0.001) and lower neonatal birth weight (*p* < 0.001) compared to the controls. No significant differences were found in terms of maternal age and parity among the two groups.

### 3.2. LMWH Increases D6 Expression of Trophoblast Cells from PE

Consistently to our previous observations [7], Western blot analysis of primary trophoblast cells lysates showed increased expression of the D6 decoy receptor in PE placentae compared to those from normal pregnant women in basal conditions (Figure 1A,B). After incubation of primary trophoblast cells with enoxaparin for 24 h (1 or 10 UI/mL), a significant increase of D6 expression was observed in trophoblast cells from PE compared to controls (Figure 1A,B). No significant modification of D6 decoy receptor expression was observed in trophoblast cells from normal pregnant women after incubation with enoxaparin.

Confocal analysis of D6-related fluorescence confirmed a significant increase of D6 expression in trophoblast cells from PE cases after treatment with enoxaparin (Figure 1C–E). The D6 decoy receptor showed an intra-cytoplasmic distribution in PE cells treated with enoxaparin, with a granular pattern, suggesting an endosomal localization (Figure 1D2). Consistently with results obtained in the Western blot analysis, no significant differences in terms of D6 expression were observed in trophoblast cells from normal pregnancy with or without LMWH treatment (Figure 1C1,C2,E). Furthermore, normal trophoblast cells showed a more diffuse and finer intra-cytoplasmic D6 distribution.

### 3.3. LMWH Restores Cytoskeleton Organization of Trophoblast Cells from PE

F-actin staining of trophoblast cells, analyzed by confocal microscopy, confirmed previous evidence of a dramatic impairment of the cytoskeleton of trophoblast cells obtained from placentae of PE women and of a normal cytoskeleton in healthy pregnant women (Figure 2A1,B1,C). Significant improvement of cytoskeletal fiber alignment, assessed in terms of eccentricity, was observed in trophoblast cells from PE women after in vitro incubation with LMWH (Figure 2B1,B2,C), reaching a value comparable to control cells. The F-actin fluorescent signal was the most intense in correspondence with the cell membrane, suggesting an enhanced actin polymerization starting from the cell periphery.

## 4. Discussion

D6 is a tissue scavenger receptor that binds CC chemokines for degradation and plays a crucial role in controlling tissue inflammation [10,11]. It is expressed in the human placenta at the maternal–fetal interface, particularly on invading extravillous trophoblasts and on the apical side of syncytiotrophoblast cells [3,17].

The implantation and placentation processes are associated with a robust maternal inflammatory response throughout pregnancy [29,30]. The extent of this inflammatory response results from a delicate balance between pro-inflammatory and anti-inflammatory mechanisms, and it is critical to successful pregnancy. This delicate balance may be compromised in pregnancy-related diseases such as PE, a placenta-induced disorder characterized by an exacerbated systemic inflammatory response [3,4,5]. Indeed, higher circulating blood levels of pro-inflammatory cytokines, like CCL2, CCL7, CCL11, IL-6, IL-8, TNF-α and RANTES, have been shown in PE [3,4,5,6,7]. In a previous study, we demonstrated that the D6 decoy receptor is up-regulated in primary trophoblast cells from PE compared to controls. Indeed, after ligand engagement, D6 increases its expression on the cell surface, due to its mobilization from the intracellular pool [7]. In contrast, a reduced trophoblast D6 scavenging ability was observed in PE, associated with a dramatic disarrangement of the cytoskeleton in PE compared to the controls. Indeed, mobilization of D6 from intracellular endosomes and vesicle trafficking relies on the regulatory and motor proteins of the cytoskeleton [7,31]. Therefore, a dynamic actin cytoskeleton is required to sustain D6-mediated uptake and targeting of chemokines for degradation. Thus, in PE, an impairment of trophoblast cell cytoskeletons, because of several potential injures, like syncytiotrophoblast oxidative stress [32], might occur and affect D6 function, that is, in the meantime, overstressed by increased D6-binding chemokine secretion. The impairment of the cell cytoskeleton, that cannot allow neither chemokine degradation nor D6 receptor recycling, causes deficient regulation of the inflammatory environment at the maternal–fetal interface.

LMWH has been proposed as a possible therapy to prevent PE recurrence based on the evidence that (i) it increases metalloproteinases synthesis and trophoblast invasiveness in trophoblast cells from spontaneous miscarriage [33], and (ii) it reduces complementary anti-phospholipid antibodies (aPL) binding to trophoblast cells [34], aPL-induced complement activation and trophoblast apoptosis [35]. Based on an in vitro study demonstrating that LMWH enhances B6FS fibrosarcoma cell adhesion and migration by inducing F-actin polymerization [24], we hypothesized a possible effect of LMWH on cytoskeleton organization, and thus on D6 decoy receptor function.

Here, we demonstrated that enoxaparin: (i) increased D6 expression in trophoblast cells from PE, conferring a characteristic granular pattern to D6 intracellular distribution, which is related to its localization in endosomes, in the context of cytoskeleton-dependent intracellular trafficking; (ii) improved in vitro F-actin eccentricity in trophoblast cells obtained from PE after incubation for 24 h, restoring the cytoskeleton spatial organization.

We can speculate that LMWH might exert these positive effects by (i) increasing intracellular synthesis of D6 in PE, and (ii) enhancing actin polymerization through a FAK-dependent mechanism. Indeed, heparin has been shown to antagonize the function of integrin adhesion receptors [36], which trigger FAK clustering [37], and to influence FAK expression and activation [38,39].

More experiments are needed to confirm our preliminary data and to determine the intracellular molecular mechanisms involved in the LMWH-induced increase of D6 expression and function, as well as in cytoskeleton reorganization in trophoblast cells from PE.

In conclusion, this study might identify, for the first time, a new beneficial mechanism of action of LMWH on placental tissues in PE. Indeed, heparin might not only exert its therapeutic effect in PE through its anti-thrombotic and pro-invasive properties, but also by improving expression and function of the anti-inflammatory D6 scavenger decoy receptor.

## Figures and Tables

**Figure 1 cells-11-02036-f001:**
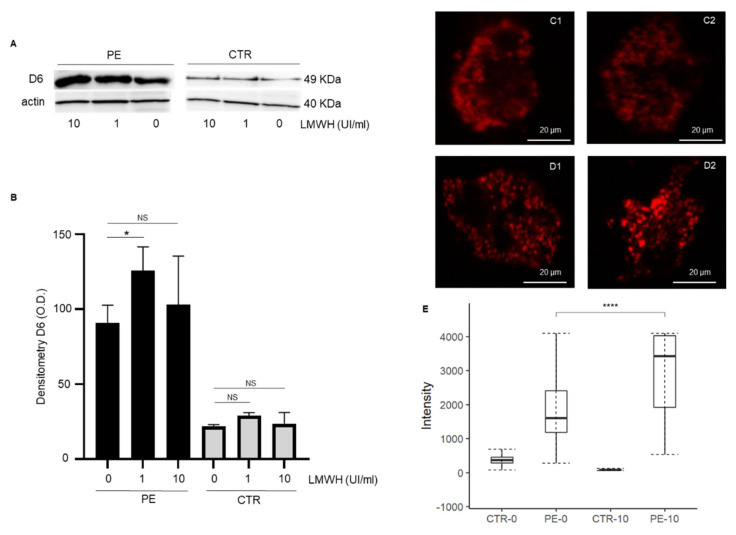
(**A**) Representative immunoblotting showing expression of D6 receptor in trophoblast cell lysates from PE or controls in basal conditions (0) and after incubation with LMWH for 24 h (1 and 10 UI/mL). (**B**) Quantitative analysis of Western blot results shows higher expression of D6 receptor in trophoblast cells from preeclamptic women (*n* = 3) after incubation with enoxaparin for 24 h (1 UI/mL) compared to basal conditions (0). No significant differences in terms of D6 expression were observed in trophoblast cells from normal pregnancy (CTR, *n* = 3) with or without LMWH treatment. (**C**–**E**) Confocal analysis of trophoblast cells showed a basal higher expression of D6 in preeclamptic women (*n* = 9) compared to controls (*n* = 9) and a significant increase of D6 expression in PE after cells incubation with LMWH (10 UI/mL) for 24 h. Data are expressed as the mean ± standard error (SE). PE: preeclampsia; CTR: control; LMWH: low molecular weight heparin; NS: not significant. * *p* < 0.05; **** *p* < 0.0001.

**Figure 2 cells-11-02036-f002:**
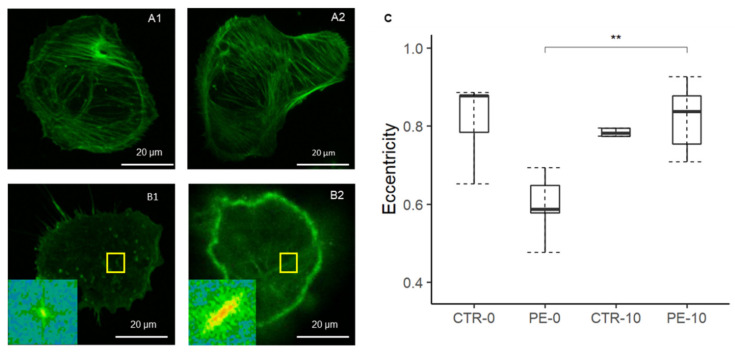
(**A1**–**B2**) F-actin staining of trophoblast cells from normal pregnant (**A1**,**A2**) and PE (**B1**,**B2**) women before (**A1**,**B1**) and after (**A2**,**B2**) LMWH incubation (10 UI/mL) for 24 h. (**C**) F-actin polymerization analyzed as eccentricity is significantly reduced in PE compared to controls. Incubation of trophoblast cells with LMWH significantly improved F-actin polymerization and cytoskeleton spatial organization. No significant difference was found in cells obtained from normal pregnant women after treatment with LMWH. Data are expressed as median with related confidence interval. ** *p* < 0.01.

**Table 1 cells-11-02036-t001:** Characteristics of patients enrolled in the study.

Characteristics	Preeclampsia (*n* = 9)	Controls (*n* = 9)	*p* Value
Age (years)	33 (21–42) **	33 (21–44) **	0.63
Smokers	1 (10%)	0 (0.0%)	0.84
Nulliparous	6 (60.0%)	5 (50.0%)	0.09
BMI at booking (Kg/m^2^)	26.3 (6.8) *	21.6 (3.2) *	<0.001
Gestational age at PE onset (weeks)	31 (19–40) **	N/A	
PE onset		N/A	
Early (<32 weeks)	7 (70.0%)		
Late (>32 weeks)	3 (30.0%)		
Gestational age at delivery (weeks)	32 (22–41) **	40 (37–41) **	<0.001
Birth weight (g)	1597 (844) *	3401 (365) *	<0.001
Birth weight percentile	24 (24) *	58 (21) *	<0.001

* Mean (standard deviation). ** Median (minimum–maximum); BMI: body mass index; PE: preeclampsia; N/A: not applicable.

## Data Availability

The data that support the findings of this study are available from the corresponding author upon reasonable request.

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
