# Peer review of "Enoxaparin Increases D6 Receptor Expression and Restores Cytoskeleton Organization in Trophoblast Cells from Preeclampsia"

_cells, 2022, doi:10.3390/cells11132036_

Round 1
Reviewer 1 Report
This study presents substantial data regarding the pathomechanism of PE and the role of heparin LMWH in the protection of this complication.
However, this preliminary study was performed on a small group of patients but I find it very interesting and relevant for our knowledge of potential mechanisms leading to PE. More experiments are needed to confirm this preliminary data and to determine the intracellular molecular mechanisms involved in LMWH-induced increase of D6 expression and function as well as in cytoskeleton re-organization in trophoblast cells from PE. But in the reviewer's opinion, this study may open further studies in that field and therefore is worthwhile to be published.
Author Response
We thank the Reviewer for his/her Kind comments and scientific evaluation of our work.
Reviewer 2 Report
The manuscript is aiming to investigate the effect of the drug "Enoxaparin" belongs to a class of drugs known as “low molecular weight heparin" on the function of D6 receptor in the context of preeclampsia.
Comments:
1-Western blotting ( 2.3): It is not clear if the author used total cellular proteins or membrane fraction of the cells for western blotting .No detergents were mentioned to solubilize the cells. why?
2- The gestational age of the PE is significantly different from the control of normal pregnancy. I wonder if the effect of Enoxaparin is dependent in the gestational age? Is there any differences in the tested parameters? D6 expression? actin organization??
3- Figure 1B- Why the analysis is based on n=3 samples and not the n=9 of control and PE samples??
4- Figure 2 C: what is the sample no. used in figure figure 2C.
5- Using 1UI/ml significantly increased the expression of D6 in trophoblast What is the consideration of using 10 UI/ml instead of 1UI/ml in the experiments presented in figure 1E and 2C ?
6- Lane 222: Figure 2 B1-D2 and C) correct B1-B2 . There is no part D2 in figure 2
Author Response
We thank the Reviewer for his/her precious comments.
- We have performed Western blotting on total cells lysates and used NP detergent during homogenization. We have edited methods according to the Reviewer’s comments.
- The gestational age at delivery of PE cases is usually Lower compared to normal pregnancies because of the common occurrence of medically-indicated preterm birth. We could not match cases and controls for gestational age for placenta collection because of ethical issues. However, among PE cases, we could not report any differences in terms of enoxaparin effect in vitro on D6 expression or cytoskeleton organization related to different gestational ages. Furthermore, We are not aware about any available evidence in the literature reporting different effects of enoxaparin on D6 expression or cytoskeleton organization in placental cell cultures.
- Western blotting has been performed only on 3 cases and 3 controls incubated with 3 different treatments (0, 1, 10 UI/ml of enoxaparin) as a confirmation experiment of confocal microscopy analysis results.
- In figure 2C there is shown the median of results from experiments of confocal analysis performed on all 9 PE cases and 9 controls.
- The concentration of 1 IU/ml in vitro has been chosen calculating the expected concentration in vivo of 4000-6000 IU diluited in about 5000 ml of plasma volume in a pregnant women. 10 UI/ ml in vitro has been tested to confirm a dose dependent and, thus, specific effect of exoxaparin in vitro.
- Thank You for the suggestion. We have edited the manuscript according to.
This manuscript is a resubmission of an earlier submission. The following is a list of the peer review reports and author responses from that submission.